# On end-to-end 6DOF object pose estimation and robustness to object scale

## 1 Reproducibility Summary

### 2 Scope of Reproducibility

3 This report contains a set of experiments that seek to reproduce the claims of two recent works related to keypoint
4 estimation, one specific to 6DOF object pose estimation, and the other presenting a generic architectural improvement
5 for keypoint estimation but demonstrated in human pose estimation. More specifically, in the backpropagatable PnP [1],
6 the authors claim that incorporating geometric optimization in a deep-learning pipeline and predicting an object's pose
7 in an end-to-end manner yields improved performance. On the other hand, HigherHRNet [2] introduces a novel heatmap
8 aggregation method that allows for scale-aware pose estimations, offering higher keypoint localization accuracy for
9 small scale objects.

### 10 Methodology

11 We used the publicly provided code where available, adapting it to fit into a model development kit to facilitate our
12 experiments. We used a dataset fit for validating both claims simultaneously, and designed a set of experiments based
13 on the published methodologies, but also went beyond seeking to validate the higher level concepts. Our experiments
14 were conducted on a Nvidia 2080 12 GB GPU with an average training time of 14 hours.

### 15 Results

16 We reproduce the claims of both papers by conducting several experiments in the UAVA dataset [3]. The integration of
17 a differentiable geometric module within an keypoint-based object pose estimation model improved its performance in
18 metrics. We additionally verify that this is the case for other differentiable PnP implementations (*i.e.* EPnP). Further,
19 our results indicate that indeed HigherHRNet improves keypoint localisation performance on small scale objects.

### 20 What was easy

21 Both papers provided publicly available implementations. In addition, many different variations were also found online.
22 Finally, the papers themselves were very clearly written, offering insights on various important details.

### 23 What was difficult

24 The main issue that required more effort was identifying the appropriate weights for BPnP [1] in order to balance the
25 different optimization objectives. As expected, this varies for the context that it is applied (task, dataset) and the values
26 presented in the paper did not work in our case. Sub-optimal selection of weights leads to convergence issues.

### 27 Communication with original authors

28 We communicated with the authors of [1] through GitHub, and we would like to thank them as they provided a fast
29 and detailed response. Furthermore, their responsiveness to past issues had already provided a nice knowledge base
30 regarding reproduction.

# 1 Introduction

Object pose estimation seeks to determines the 3D position and orientation of an object in camera-centred coordinates. During the last years, two main directions have been emerged for data-driven 6DOF object pose estimation; *direct pose regression* which predict pose in an end-to-end manner, and *indirect* that learns the surrogate task of keypoint localisation and then solves a Perspective-n-Point (PnP) problem to estimate the resulting pose. Even though it has been shown [4] that the latter methods better approach the problem, there are still open challenges that need to be solved. One issue is the splitting between the actual task at hand, and the surrogate task that they learn. The other has to do with the spatial nature of keypoint localisation and smaller scale objects. Recently, two works have been presented that seek to address these issues, BPnP [1] and HigherHRNet [2]. In this work, we seek to reproduce and verify their claims in a task that is relevant for both of these works, drone pose estimation. While BPnP's relation has to do with the task at hand, HigherHRNet is also relevant because commodity drones are usually small form objects, and when flying around the further distance themselves from the operator, effectively reducing their scale in the camera's image domain.

# 2 Scope of reproducibility

Consequently, we opt for reproducing the claims of both of these two relevant papers addressing the aforementioned challenges. In more details, the authors of BPnP [1] propose a novel differentiable module which calculates the derivatives of a PnP solver through implicit differentiation, enabling the backpropagation of its gradients to the network parameters, and as such allowing for end-to-end optimization and learning. On the other hand, the authors of HigherHRNet [2] focus on improving the 2d landmarks' localization performance for smaller-scale humans by proposing a novel multi-scale supervision scheme for training and a heatmap aggregation module for inference.

The main claims of both papers can be summarised below:

- **BPnP:** An end-to-end trainable pipeline for object pose estimation, can achieve greater accuracy by combing the reprojection losses (Table 3).
- **HigherHRNet:** A novel method for learning scale-aware representations using high-resolution feature pyramids, eventually achieving greater results for small scale objects[1] (Table 4).

# 3 Methodology

We implemented our experiments by re-using the publicly available implementation for BPnP, and implementing HigherHRNet after styding the paper, the original publicly available implementation, as well as other implementations. In both cases we integrated the code base in a modular framework that facilitates reproducible experiments [5], which generally required slight modifications of the original code provided by the authors to fit its requirements. The overall methodology for our experiments is depicted in Figure 1. On the left, a traditional monocular heatmap-based keypoint localisation pipeline is presented, whereas on the right, the BPnP required components are illustrated.

**BPnP:** BPnP focuses on the *Pose Retrieval stage*, and following [1] we trained our model under the 3 different schemes used in the original work as well:

- heatmap loss ($l_h$),
- mixture loss $l_m = l_h + \beta * l_{proj}$,
- and pose loss $l_p = l_{reg} + l_{proj}$,

where $l_{proj} = \|\pi(z|y, K) - x^*\|_2^2$ and $l_{reg} = \|x - \pi(z|y, K)\|_2^2$. Also, $\pi$ is the projection function employing the predicted pose($y$) from the PnP solver, the corresponding object's 3D points $z$ and $K$ the camera intrinsic matrix. Apart from these experiments presented also in the original paper, we conducted an extra set of experiments that aimed at validation the concept of end-to-end 6DOF pose estimation via differentiable PnP. We used another openly available differentiable PnP implementation, and additionally, also tested the faster counterpart of BPnP. We present results across many established object pose estimation metrics, as well as computational performance metrics for all the aforementioned experiments.

**HigherHRNet:** On the other hand, for HigherHRNet we focused on the *Heatmap regression part* by using different models for the decoder part of the architecture, with details following in Section 3.1.

---

[1]We apply the proposed module in the object pose estimation task, while authors originally demonstrated it for the human-pose estimation task, but its concept still applies in our case as well.

76 All the code and its documentation are submitted and published along with this report.

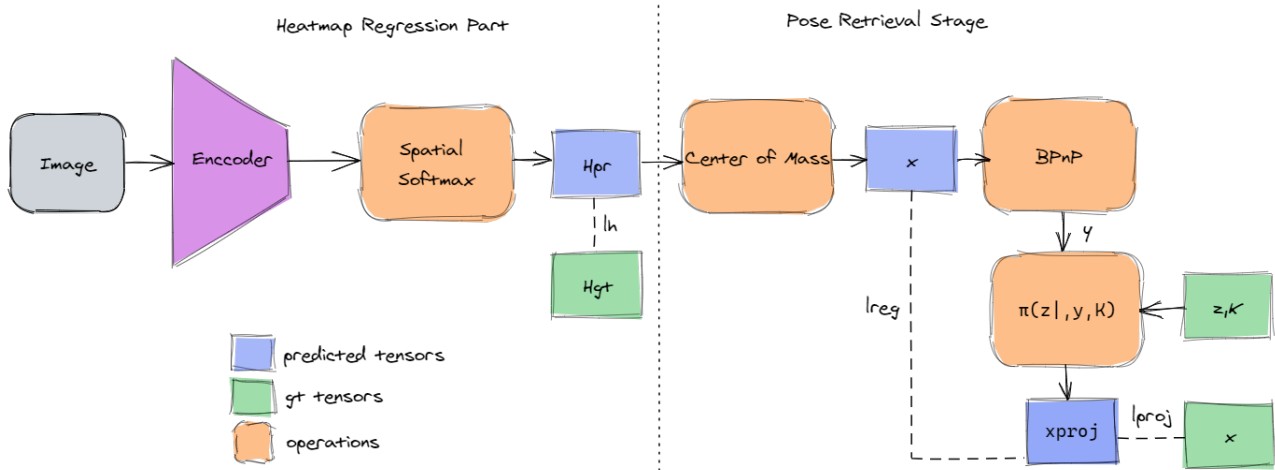

Figure 1: Indirect object pose estimation approach consisting of the *Heatmap Regression* part where HigherHRNet paper focuses on, and *Pose Retrieval* part where BPnP focuses, alongside the supervision signals.

## 3.1 Model descriptions

78 The following models were used as our backbone for regressing the heatmaps and the corresponding coordinate spatial
79 distributions (the decoder part in Figure 1):

80 • *HRNet [6]* with feature maps of width 48 and 3 stages. The 2nd, and 3rd, stages contain 1, 4 exchange blocks,
81 respectively, and each exchange block contains 4 residual units.

82 • *HigherHRNet [2]* with feature maps of width 48 and 3 stages. The 2nd, and 3rd, stages contain 1, 4 exchange
83 blocks, respectively, and each exchange block contains 4 residual units.

84 • *Stacked Hourglass [7]* with depth 2 and feature maps of width 128

85 Following the heatmap predictions, we apply a decoding operation (*i.e.* center of mass specifically) in order to extract
86 the keypoints from the heatmaps, which are later driven to the PnP algorithm for retrieving the 6D pose.

87 In addition, we integrated the EPnP [8] algorithm, using the available implementation in Pytorch3D [9] instead of
88 BPnP to assess whether other – similar in concept – implementations can verify the claim and quantify the differences
89 between these approaches. As a side-note, it should be mentioned that our models' configuration slightly differs from
90 the ones described in the original works in order to comply with the working resolution of the dataset we used.

## 3.2 Datasets

### 3.2.1 UAVA

93 As aforementioned, our experiments were conducted on a dataset that allowed for the validation of both works
94 simultaneously. This also better helps in deducing whether the claims are reproducible as the context (task or dataset)
95 can vary. We used the UAVA dataset[2] for object pose estimation. UAVA targets human-robot cooperative Unmanned
96 Aerial Vehicle(UAV) applications and offers two different drone models, namely DJI M2ED[3] and Ryze Tello[4]. The
97 UAVA dataset provides the 3D models of both drones accompanied by ground-truth annotations such as 3D bounding
98 boxes, 6D pose, and at the same time multi-modal data. More importantly, the difference in size between the two drone
99 models allows for the validation of scale-invariant pose estimation.

---

[2] https://vcl3d.github.io/UAVA/
[3] https://www.dji.com/gr/mavic-2-enterprise
[4] https://www.ryzerobotics.com/tello

### 3.2.2 Preprocessing

We processed the original dataset in order to keep only the samples where all the 2D keypoints are within the image, given that BPnP relies on softly approximating the coordinate, and that would fail in the case of out of field-of-view keypoints. However, we should mention that we did not apply any other filtering (i.e. visibility of all the keypoints, boundary cases, etc.).

### 3.3 Hyperparameters

We train all the models for **44** epochs and select the best performing model for testing. We used the **Adam** optimizer with a learning rate of $1e-4$, betas of values $0.9$ and $0.999$ and no weight decay, and a seed value of $1989$ for ensuring reproducibility. Albeit, we experimented with different losses (*i.e.* KL, MSE) for $l_h$, we found that L1 loss works the best, offering the best results and faster convergence. This could be attributed to the different resolution of the heatmaps grid (in our case is lower) as well as the different configuration of the heatmap decoder model (we used 3 stages instead of 4). It is worth mentioning that we also tried a bigger heatmap resolution (*e.g.* $160 \times 120$) although we decided to conduct our final experiments in the lower resolution for two main reasons. First, most heatmaps regression decoders used in the literature make their prediction in the $1/4$ of the original image, and second, this higher heatmap resolution would enforce us to further reduce the depth of the decoder model. Specifically, for BPnP we set $\beta$ value to $1e-5$ after conducting a greedy heuristic search, with values ranging from $0.001$ to $1e-9$, as the proposed value for $\beta$ coefficient, did not work for our case. The selection of a non-appropriate $\beta$ coefficient value can lead to stability issues as noted in Section 5.2.

### 3.4 Experimental setup and code

As mentioned above, we integrated the authors' code (BPnP) or our own re-implementations (HigherHRNet) in [5] which is a PyTorch framework for modular and reproducible workflows[5]. Each model is implemented in a configuration file that defines the different components (optimizer, datasets, model architecture, pre-/post-processing graphs, etc.) and logs all hyperparameters. For each experiment we report the standard metrics below:

**NPE**: is the magnitude (L2-norm) of difference between the ground-truth and estimated position vectors from the origin of the camera reference frame to that of the drone body frame, normalised with ground-truth vector.

**AD**: is the angular distance between the predicted, rotation matrix, and ground-truth, or in other words, the magnitude of the rotation that aligns the drone body frame with the camera reference frame.

**ACC**: considers an estimated pose to be correct if its rotation error is within **k°** and the translation error is below **k cm**.

**ADD**: is the average distance metric to compute the averaged distance between points transformed using the estimated pose and the ground truth pose. Eventually, a pose estimation is considered to be correct if the computed average distance is within **k%** of the model diagonal.

**Proj**: is the mean distance between 2D keypoints projected with the estimated pose and those projected with ground truth pose. An estimated pose is considered correct if this distance is within a threshold **k**.

### 3.5 Computational requirements

Table 1 showcases the total duration of each experiment (with a 24 batch size) as well as some other useful statistics such as the mean duration time for a forward pass, a backward pass, an optimizer step, as well as the total test duration with batch size 1. It is clear, that the introduction of the differentiable PnP modules in the training procedure increases the total training time significantly, as the backward and step operation require more time. We ran our experiments on a machine with the specifications presented in Table 2.

## 4 Results

Our results support the claims presented by both authors in [1] and [2] respectively. As is demonstrated in Table 3, the model trained with $l_p$ achieved better results in most of the metrics for both drone models. Similarly, Table 4 indicates that HigherHRNet yields better results for the small-scale drone in most of the metrics, although its performance for the bigger M2ED drone is worse compared to the standard HRNet model.

---

[5] www.github.com/ai-in-motion/moai

Table 1: Time statistics for each experiment. Red and orange colors indicate the two (worst, and second worst respectively) most time-consuming experiment per drone model.

| Drone | | Total Training Duration (hrs) | Mean Model Fwd Duration(s) | Mean Model Bwd Duration(s) | Mean Optimizer Step (s) | Total Test Duration (min) |
|---|---|---|---|---|---|---|
| M2ED | $l_m$ | 14.14 | 0.13 | 2.73 | 2.89 | 23.75 |
| | $l_p$ | 14.13 | 0.13 | 2.73 | 2.90 | 19.30 |
| | EPnP | 17.33 | 0.19 | 1.49 | 1.74 | 24.37 |
| | HRNet | 11.19 | 0.06 | **0.003** | **0.21** | 19.87 |
| | Hourglass | **6.99** | 0.14 | 0.019 | 0.22 | 15.85 |
| | HigherHRNet | 10.54 | 0.11 | 0.033 | 0.36 | 23.63 |
| Tello | $l_m$ | 20.78 | 0.13 | 2.72 | 2.9 | 23.82 |
| | $l_p$ | 20.38 | 0.13 | 2.68 | 2.85 | 19.95 |
| | EPnP | 21.80 | 0.28 | 3.80 | 4.14 | 19.13 |
| | HRNet | **9.69** | 0.13 | 0.029 | 0.38 | 19.88 |
| | Hourglass | 10.13 | 0.14 | **0.020** | **0.22** | 15.62 |
| | HigherHRNet | 16.10 | 0.14 | 0.032 | 0.4 | 20.50 |

Table 2: Hardware Components

| | |
|---|---|
| *OS* | Windows Microsoft Pro (x64) |
| *Storage* | 3TB Toshiba HDD |
| *CPU* | Intel i9-7900X (4.30 GHz) |
| *GPU* | GeForce RTX 2080 Ti (12 GB) |
| *RAM* | 4 x 16 GB Kingston (2666 MHz) |

## 4.1 Results reproducing original papers

### 4.1.1 BPnP

With these experiments we show that the addition of a differentiable PnP module improves the performance in object pose estimation task. We provide qualitative results in Figure 3. It is worth highlighting that training with $l_p$ does not restrict the shape of the distribution the way that it is constrained when relying on heatmap supervision (*i.e.* Gaussian distribution approximation). Instead, the model freely localizes the keypoints, which results in more focused predictions. This is illustrated in Figure 2 where qualitative results display the heatmaps on top of the color images.

Table 3: BPnP results on the UAVA dataset. We trained all models for 44 epochs and select the best among them for inference. Light green with **bold** and light blue indicate the best and second best performers.

| Drone | | NPE↓ | AD↓ | ACC2↑ | ACC5↑ | ADD2↑ | ADD5↑ | Proj2↑ | Proj5↑ |
|---|---|---|---|---|---|---|---|---|---|
| M2ED | $l_m$ | 0.014 | 0.027 | 92.13 | 98.07 | 81.31 | 93.34 | **99.45** | **99.58** |
| | $l_p$ | 0.012 | 0.026 | **95.20** | **98.36** | **90.29** | **96.81** | 98.05 | 99.14 |
| | $l_h$ | **0.011** | **0.020** | 90.75 | 98.04 | 80.52 | 91.37 | 97.56 | 99.49 |
| Tello | $l_m$ | 0.071 | **0.189** | 43.38 | 82.11 | 14.88 | 41.47 | 93.97 | **96.08** |
| | $l_p$ | **0.063** | 0.223 | **55.31** | **85.34** | **20.04** | **50.53** | **93.19** | 94.49 |
| | $l_h$ | 0.091 | 0.252 | 36.36 | 74.99 | 18.27 | 43.31 | 89.25 | 94.00 |

### 4.1.2 HigherHRNet

These experiments showcase that the addition of the aggregation module improves the keypoint localization performance when targeting smaller-scale objects. Specifically, the HigherHRNet architecture gives better results in most of the metrics for the small form drone (Tello). On the other hand though, this is not the case for the larger drone, where the HigherHRNet performance is slightly worse than the standard HRNet's one.

## 4.2 Results beyond the BPnP paper

Apart from the experiments conducted by the authors in [1] we provide additional to further support the main claims. Particularly, we compared BPnP versus an alternative differentiable PnP algorithm (*i.e.* EPnP) and the results are

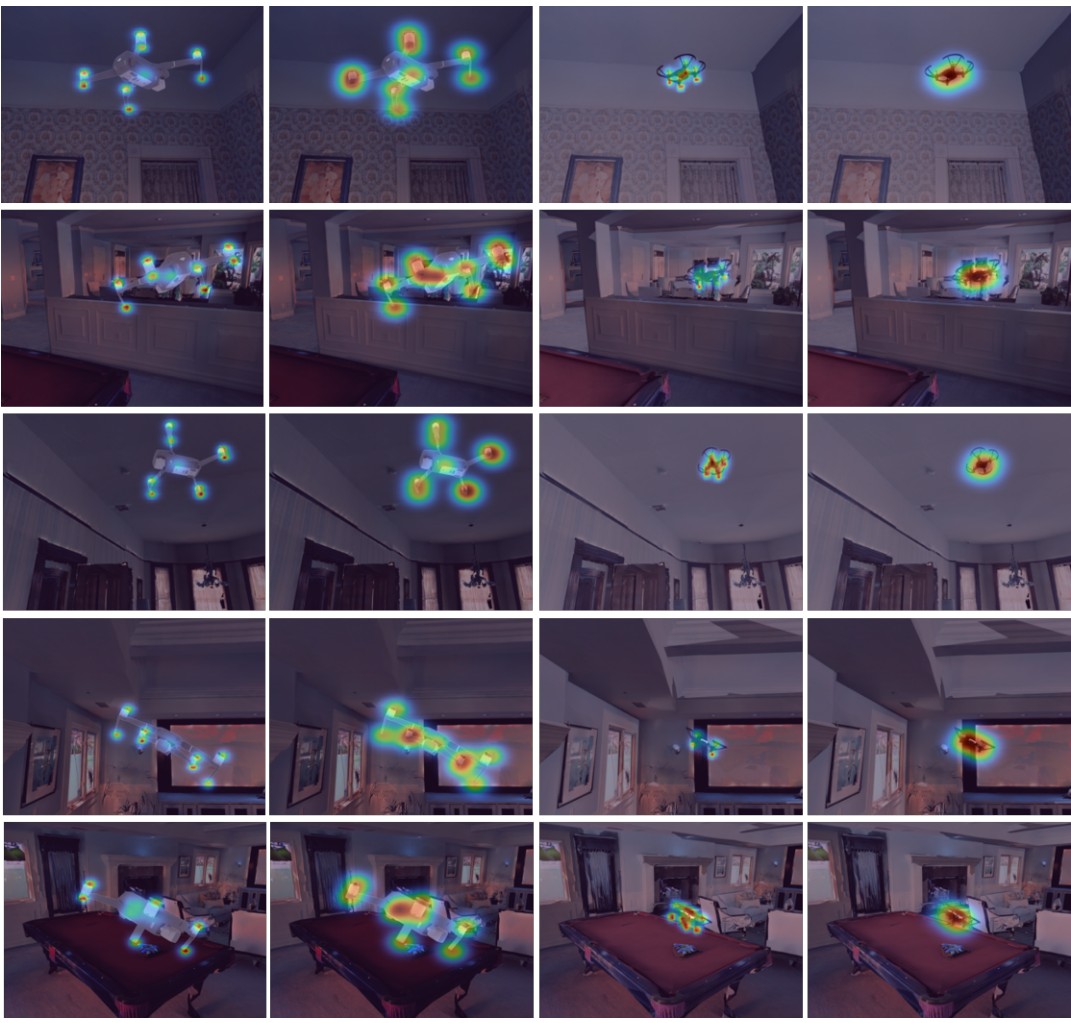

Figure 2: Qualitative heatmap results on random samples from the test-set. The first **two** columns depicts the M2ED drone while the last **two** the Tello drone; with heatmaps predicted by models trained with $l_p$ and $l_h$ respectively. Evidently, heatmaps by the $l_h$ model (**second and fourth columns** tend to keep the 2D Gaussian shape distribution, while the $l_p$ ones (**first and third columns**) freely approximate the $(x, y)$ position enforced by the regularizer term($l_{reg}$) of the $l_p$ loss, eventually allowing for more distinguishable 2D keypoints estimations.

Table 4: HigherHRNet results on the UAVA dataset. We trained all models for 44 epochs and select the best among them for inference. Light green with **bold** and light blue indicate the best and second best performers.

| drone | | NPE↓ | AD↓ | ACC2↑ | ACC5↑ | ADD2↑ | ADD5↑ | Proj2↑ | Proj5↑ |
|---|---|---|---|---|---|---|---|---|---|
| | Hourglass | 0.015 | 0.028 | 89.43 | 96.94 | 78.20 | 90.42 | 96.56 | 99.02 |
| *M2ED* | HRNet | **0.011** | **0.020** | **90.75** | **98.04** | **80.52** | **91.37** | **97.56** | **99.49** |
| | HigherHRNet | **0.011** | **0.020** | 89.92 | 97.75 | 79.58 | 90.99 | 97.50 | 99.44 |
| | Hourglass | 0.094 | **0.214** | 32.19 | 75.47 | 14.76 | 38.26 | **92.23** | **96.25** |
| *Tello* | HRNet | **0.091** | 0.252 | 36.36 | 74.99 | 18.27 | 43.31 | 89.25 | 94.00 |
| | HigherHRNet | 0.095 | 0.264 | **42.98** | **75.69** | **20.19** | **46.69** | 89.54 | 93.63 |

159 demonstrated in Table 5. We also provide extra experiments of a BPnP implementation in which the calculation of the
160 higher-order derivatives is ignored from the coefficient's graph as presented in Table 6.

### 4.2.1 BPnP vs EPnP

For this experiment we utilised the same backbone (*i.e.* HRNet) but we changed the BPnP module with the EPnP. We followed the exact same training procedure, hyperparameters, as well as the same loss $l_m$. Results are summarized in Table 5. It is evident that EPnP and BPnP offers comparable results in most of the metrics.

Table 5: BPnP vs EPnP. Following the same approach we trained the decoder part only with $l_h$ for 30 epochs and then continue with $l_m$ for 14 epochs. Light green with **bold** indicates the best performer.

| Drone | | NPE↓ | AD↓ | ACC2↑ | ACC5↑ | ADD2↑ | ADD5↑ | Proj2↑ | Proj5↑ |
|---|---|---|---|---|---|---|---|---|---|
| M2ED | BPnP | **0.014** | **0.027** | 92.13 | 98.07 | **81.31** | 93.34 | 99.45 | 99.58 |
| | EPnP | **0.014** | **0.027** | **92.89** | **98.19** | 80.50 | **93.64** | **99.52** | **99.61** |
| Tello | BPnP | **0.071** | **0.189** | 43.38 | **82.11** | 14.88 | 41.47 | **93.97** | 96.08 |
| | EPnP | 0.074 | 0.192 | **46.77** | 81.64 | **21.07** | **49.59** | 93.77 | **96.13** |

### 4.2.2 BPnP$_{faster}$

Authors in [1] provided an alternative method for calculating the gradients through the PnP layer, which essentially is the same method as the original, although ignoring the higher-order derivatives from the coefficients graph. Therefore, we provide results using this faster BPnP method in Table 6, comparing the two different versions, as well as their training times in Table 7. It seems that the original version outmatches the faster one, albeit there is no significant performance drop. On the other hand, Table 7 indicates how the second implementation justifies its name. So, it is in users' fluency whether they need to sacrifice gradient accuracy and some performance drop in exchange for efficient training times.

Table 6: BPnP$_{faster}$ results on the UAVA dataset, following the exact training approach as original BPnP. Here we present results with models trained with $l_p$. Light green with **bold** indicates the best performer.

| Drone | | NPE↓ | AD↓ | ACC2↑ | ACC5↑ | ADD2↑ | ADD5↑ | Proj2↑ | Proj5↑ |
|---|---|---|---|---|---|---|---|---|---|
| M2ED | BPnP$_{faster}$ | 0.013 | 0.029 | 94.79 | 98.06 | 89.38 | 96.66 | 97.83 | 98.98 |
| | BPnP | **0.012** | **0.026** | **95.20** | **98.36** | **90.29** | **96.81** | **98.05** | **99.14** |
| Tello | BPnP$_{faster}$ | **0.055** | **0.167** | **55.42** | **87.03** | **26.43** | **58.99** | **94.91** | **96.12** |
| | BPnP | 0.063 | 0.223 | 55.31 | 85.34 | 20.04 | 50.53 | 93.19 | 94.49 |

Table 7: BPnP$_{faster}$ time statistics. Light green with **bold** indicates quicker performance.

| Drone | | Total Training Duration (hrs) | Mean Model Fwd Duration(s) | Mean Model Bwd Duration(s) | Mean Optimizer Step (s) | Total Test Duration (min) |
|---|---|---|---|---|---|---|
| M2ED | BPnP$_{faster}$ | **6.80** | 0.089 | **0.53** | **0.66** | 17.15 |
| | BPnP | 14.13 | 0.13 | 2.73 | 2.90 | 19.30 |
| Tello | BPnP$_{faster}$ | **10.39** | 0.09 | **0.54** | **0.69** | 18.85 |
| | BPnP | 20.38 | 0.13 | 2.68 | 2.85 | 19.95 |

## 5 Discussion

After conducting several experiments on the UAVA dataset, the central claims of [1] and [2] stand true; as they both outperform other methods. Particularly, for validating BPnP we conducted the same experiments as the original paper, and further, we compare it with another differentiable PnP method (*i.e.* EPnP). The inclusion of 2D-3D geometry constraints through differentiable geometric optimization, improves the performance. Extending the experiments of the original paper, we compare another implementation of the BPnP module which ignores the high order derivatives from the coefficient graph. This module achieves comparable results as its counterpart apart it is much faster. It is worth noting, that both BPnP, and EPnP are quite time-consuming as demonstrated in Table 1. Finally, we study the performance of the HigherHRNet [1] in a very challenging small scale object. Indeed, the performance of the proposed heatmap aggregation module achieves better results when compared with other well-established methods.

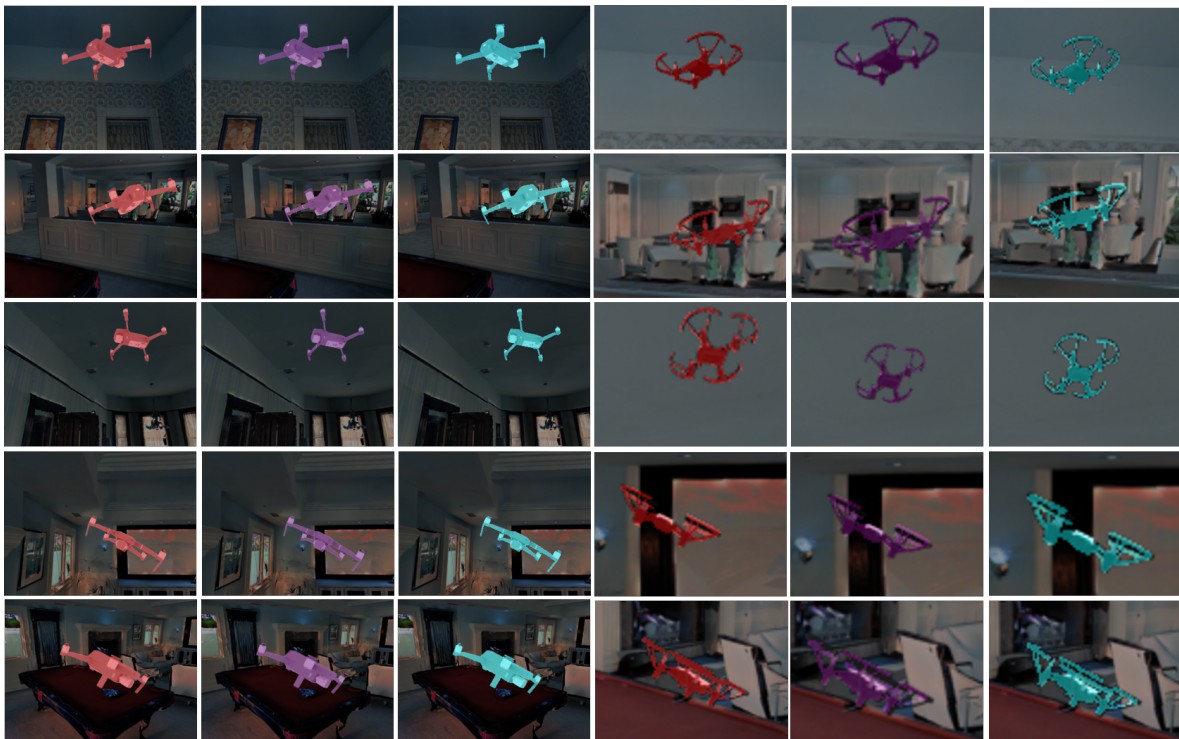

Figure 3: Qualitative results on random samples from the UAVA dataset from three different models. The **red** mask indicates predictions by $l_p$ trained model, **purple** by $l_m$ and finally **cyan** by HRNet. The drone masks are rendered by employing the predicted pose(i.e. output of the BPnP) and then blended with the original color image. The first **three** columns depicts M2ED drone model while the rest **three** the Tello drone. The Tello samples are cropped and zoomed-in due to its small form factor.

## 5.1 What was easy

Implementing most of the code was straightforward as authors of both papers provide source code. GitHub issues were another source of retrieving information, clarifying parts of the papers when needed. Additionally, both of the original papers are quite complete, well-written making it easy to follow.

## 5.2 What was difficult

Our major difficulty was related to finding the appropriate value for balancing the terms of mixture loss $l_m$, aka the $\beta$ value. Even though, authors in [1] provided the value that they used for their experiments this did not work for us, as this is a case specific parameter. It is worth noting that a non-appropriate selection of the balancing term can lead to convergence issues and negative results. Even though, not related with the code of both of the papers, we feel that it would be constitutive to mention that we faced the same difficulties when trying to incorporate EPnP in our workflow.

## 5.3 Communication with original authors

Authors of [1] did not specify the configuration of the used network in the pose estimation task, nor the hyperparameters. Thus, we contacted them through GitHub where they provided a detailed answer, available now to the research community. We did not contact HigherHRNet authors [2] as the online implementations and the text and figures in their paper were a good enough guide to understand and implement it.

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
