# OpenReview forum: "On end-to-end 6DOF object pose estimation and robustness to object scale"
_ML_Reproducibility_Challenge/2020 — RC2020_

### Official Review · AnonReviewer3 · 2021-02-25
**BPnP for end-to-end 6DOF object pose estimation is a reproducible result**

**Rating:** 7
**Confidence:** 4

**Review:**

**Reproducibility Summary**:
The authors have provided a detailed summary meeting the requirements.


**Scope of reproducibility**:
Yes, the reproducibility report has clearly and concisely stated the scope of reproducibility.

**Code**:
Yes, the authors have re-used the original author's code repository and also tried with another differential PnP (i.e EPnP) module as described in the reproducibility report.

**Communication with original authors**
Yes, the authors connected with one of the BPnP paper's original authors through their Github repo.   The authors did not reach out to HigherHRNet paper's original authors.

**Hyperparameter Search**:
Yes, the authors have attempted to reproduce the hyperparameter search, but the $\beta$ coefficient from the original author's (BPnP)'s code did not work for the authors of the reproducibility report.

**Ablation Study**:
Yes, the authors used an alternative implementation of the BPnP module to review the results and reproducibility.  The authors tried ignoring the higher-order derivatives of the BPnP.

**Discussion on results**:
Yes, the reproducibility report contains a brief discussion on the state of reproducibility of the original papers, but does not highlight which parts are easy to reproduce or which parts were harder.  They could have mentioned the difficulty with the $\beta$ parameter here.

**Recommendations for reproducibility**:
No, the authors did not provide any recommendation to the original authors for improving reproducibility.

**Results beyond the paper**:
The authors have tried additional differentiable PnP implementation (EPnP) to verify the claim.  That is a good point.  Another good point is that the authors tried to reduce the complexity and run time of the model using a faster BPnP, then evaluated the results and provided the detail pros and cons of using the technique; bonus point to the authors for that.  The authors include significantly more quantitative and qualitative results than the original paper.

**Overall organization and clarity**:
Nicely written and coherent.

**Pros**:
Significantly more quantitative and qualitative results.

**Con**:
The authors highlight the best results in the tables using a red color.  A better choice would be green or yellow or just to leave it uncolored.

**Familiar With The Original Paper:**

I have read the original paper

**Reproducibility Summary:**

Report has summary

---

### Official Review · AnonReviewer2 · 2021-03-01
**results beyond the original paper**

**Rating:** 7
**Confidence:** 3

**Review:**

Two papers are reproduced here: backpropagatable PnP & HigherHRNet, for the problem of 6DOF object pose estimation. The results are evaluated in the UAVA dataset.
The work contains the mentioned points, including communication with original authors, and discussion of the reults. Overall, the results are meaningful. They even present results beyond the original paper, such as section 4.2.


**Familiar With The Original Paper:**

I have not read the original paper

**Reproducibility Summary:**

Report has summary

---

### Official Review · AnonReviewer1 · 2021-03-02
**Sensible validation of BPnP**

**Rating:** 7
**Confidence:** 4

**Review:**

The report aims to verify the effectiveness of using Backpropogatable PnP (BPNP) on a pose estimation task with drones. Following the original paper, the setup uses heatmap regression, from which the object pose is extracted and refined through PnP, given 3D geometry. The incorporation of geometric constraints (e.g. the projection loss from the BPnP [1] paper) is claimed to improve estimation of keypoints.

In addition to BPnP, the report also examines the effect of scale aggregation from the HigherHRNet paper which proposes a scheme to bottom up scheme for aggregation in stacked hourglass type setups for heat map computation. The report goes about doing this through two types of drones of varying sizes - Mavic (larger) and Tello (smaller) using the dataset UAVA which contains ground truth annotations needed (e.g. 6D pose).

BPNP scheme is compared with a reference differentiable implementation (EPnP) via PyTorch3D.

The report was well written, and the experiments thoughtfully carried out. In general, the numbers show improvements in keypoint estimation after incorporating the the projection losses. The comparison with EPnP is also quite favourable. They also show that the 'faster' version of BPnP reduces computational time without loss of accuracy.

Pros: Major ideas in paper explained clearly, with cogent implementation results. I would be keen on using BPnP for practical tasks.

Cons: Hyperparameter sweep tries not touched upon in detail. In particular, how does one chose the weighting parameters? Were any stability issues encountered? I would have liked to see a more varied list of scales as in the original paper rather than just the two drones, and (please correct me if I am not reading it correctly) the numbers are not markedly better/worse across implementations (Table 4).

[1] BPnP: https://openaccess.thecvf.com/content_CVPR_2020/papers/Chen_End-to-End_Learnable_Geometric_Vision_by_Backpropagating_PnP_Optimization_CVPR_2020_paper.pdf

**Familiar With The Original Paper:**

I have read the original paper

**Reproducibility Summary:**

Report has summary

---

### Decision · Program_Chairs · 2021-03-31

**Decision:**

Accept

**Comment:**

Selected for ReScience-C Journal Publication.